# Spaced Scheduling Enhances Instruction-Prompted Reasoning in Large Language Models

## Abstract

The recent popularity of large language models has been fueled in part by advances in instruction tuning, which has helped unlock new levels of zero-shot model performance. Much of the prior work in this area has focused on creating new datasets to improve or add specific skills (e.g., improving reasoning via chain-of-thought prompting), or improving existing data sets by increasing the diversity of tasks and prompting templates. However, studies have shown that instruction tuning can sometimes lead to performance degradation, and recent work has sought to overcome this issue by creating better dataset mixes (or collections) involving laborious and careful ablation studies to find the *right* composition. In this work, we propose a novel adaptive scheduling strategy we call *spaced scheduling* motivated by the spaced repetition learning method used by humans that creates an optimal curriculum (or schedule) of training examples. Our approach aims to perform the data mix selection process online during training, tailoring the training data composition to the chosen pre-trained model, reducing the need for extensive studies over different compositions of training data. Our results show that Spaced Scheduling yields better performance than random sampling and other pruning and scheduling baselines and comparable results in the worst case, using less training data and minimizing catastrophic forgetting. Further, our proposed approach also yields more *balanced* performance across all subcategories of the tested benchmarks.

## 1 Introduction

*Spaced repetition* (SR) is a learning technique from cognitive science that involves reviewing information at gradually increasing intervals over time, with early work on the subject dating back to Ebbinghaus (1885), and the well-known Ebbinghaus model of the *forgetting curve*. The spaced repetition methodology capitalizes on the *spacing effect* (Hintzman, 1974), a psychological principle that posits that our brains retain information more effectively when we learn in multiple, spread-out sessions. When applied to acquiring multiple skills, spaced repetition can be particularly effective. By revisiting different skills intermittently, learners can ensure that each skill gets the refreshed attention it requires just as memories start to fade. This not only aids in the retention of individual skills but also facilitates the integration of multiple skills, as the interplay between them during the learning phase can create more robust neural pathways and richer contexts. The result is improved mastery and recall, making the learner more adept and versatile across various skills.

Prior work (Amiri et al., 2017) explored applying SR to Deep Learning (DL) training and showed that the latter is affected by the same factors controlling human learning (i.e., difficulty, spacing, and memory strength). Their proposed algorithm can match the performance of standard training while using less training data, yielding a shorter training time. Whereas Amiri et al. (2017) focused on previous-generation DL models (e.g., MLP (Joulin et al., 2017), LSTM (Sutskever et al., 2014)), trained on simple (by today's standard) tasks, such as sentiment analysis in a single task-tuning setting, we are interested here in determining if some of these concepts can be adapted and applied to tune modern *pre-trained* LLMs on instruction data (Ouyang et al., 2022; Mishra et al., 2022) in a large-scale (1k+ tasks (Longpre et al., 2023)) multitask set up. As one uses SR to schedule the training examples based on a given complexity or difficulty measure, the notion of curriculum naturally

arises. The seminal work of Bengio et al. (2009) on *curriculum learning* (CL) further developed and formalized the strategy of using a curriculum in machine learning—based on ordering sequences of training examples, in a manner inspired by the ordering of tasks into increasing complexity in human education. In contrast to CL, here we propose and examine a strategy that is particularly well-suited to modern LLMs. We also explore how these strategies affect catastrophic forgetting (CE) since studies (Luo et al., 2023) that the tuning process can erase the knowledge acquired during pre-training. Existing data pruning work (Marion et al., 2023; Attendu & Corbeil, 2023) shows that training data can be reduced while keeping or outperforming the non-pruned counterparts, but highlights that the success of any pruning strategy depends on the goodness of the chosen pruning metric (Sorscher et al., 2022). Here, we are interested in exploring how SR and data-pruning strategies can work together in an online and dynamic way to optimize the instruction finetuning (IFT) process of pre-trained LLMs, especially the ones with significant zero-shot performance.

In this work, we introduce a novel DL training approach that we call Spaced Scheduled Training (SST) which extends prior work using SR to modern LLMs supplemented by a dynamic data pruning strategy where elements of CL arise as a consequence of scheduling or skipping problem examples or examples that the learner has already mastered. Our approach requires no additional analysis of task complexity to predetermine a curriculum or schedule; one starts with a standard randomized set of examples. With our approach, a model will see examples of all types of difficulty in the first epoch of training but will adaptively avoid expending computation on "easy", currently too "hard", and inherently ambiguous examples in the future.

Our contributions can be summarized as follows:

- We introduce a novel training strategy we call *spaced scheduling training* motivated by spaced repetition used by humans for optimal learning.
- Extensive empirical evaluation and careful statistical analysis show that spaced scheduling reliably increases the performance of instruction-tuned LLMs, especially on reasoning tasks.

## 2 RELATED WORK

**Spaced Repetition** is an efficient learning technique used by humans to enhance long-term information retention. This approach relies on a repeated review of content using *active recall*—recalling the information without looking at the material (e.g., flashcards). The reviews are temporarily *spaced* following a schedule based on how well one recalls the information. The spacing is usually calculated by an algorithm (e.g., SuperMemo (Wharton, 1994), MEMORIZE (Tabibian et al., 2019)) taking into account additional aspects such as the number of successful consecutive recalls. This technique was initially introduced by psychologists in the 1930s (Spitzer, 1939), and since then, there have been multiple studies exploring the approach and showing its benefits (Yuan, 2022; Karpicke & Roediger, 2008) on both the memory strength and concept-understanding abilities. Other studies showed that it is also beneficial when forming long-term memory in other species (e.g., in drosophila (Jacob & Waddell, 2020)), demonstrating that the neurological mechanisms on which this approach relies extend beyond human learning. Early human psychology study (Ebbinghaus, 2013) explored a simple but fundamental memory model–a theoretical framework that explains how human memory processes information, called *exponential forgetting curve* which states that the probability of recalling learned information if not reviewed, decays exponentially. This decay is a function of the time since the last review and the memory strength, where the latter improves as a function of the number of reviews and spacing between these reviews. Later study (Reddy et al., 2016), showed that the difficulty is also an important factor affecting the recall probability that can be written as

$$\Pr(recall) = \exp(-\frac{difficulty \times delay}{strength}) \tag{1}$$

Amiri et al. (2017) was the first to apply SR to training DL models, they showed that DL training is affected by the same three factors as human memory and introduced an algorithm called Repeat before Forgetting (RbF) that aims to focus on difficult examples. Their single-task finetuning results on sentiment classification, image categorization, and arithmetic addition, showed that RbF can reduce the data usage by up to 66% per epoch, leading to 2.9 times faster training. More importantly, this work also showed empirically that the factors affecting the recall probability in humans (i.e., the

difficulty of the example, the delay since the last review, and the memory strength) are also affecting DL models, motivating the use of SR in DL training.

**Data Pruning** is the process of filtering the training dataset of a DL model. This approach aims to remove non-useful or possibly harmful examples (e.g., mislabelled examples) without affecting the final performance which also reduces the compute requirement. For example, Marion et al. (2023) used a perplexity-based (PPL) offline method to filter the pre-training dataset of LLMs. Their proposed method uses PPL computed using an external reference model to prune examples with low values—-higher probability text. The authors showed that the simple PPL metric performs better than more complex metrics (e.g., Error L2-Norm (EL2N)). Their results showed that they can achieve the non-pruned performance with only 30% of the data. In computer vision, Sorscher et al. (2022) explored static pruning using the example's proximity to the decision boundary as a difficulty measure, where harder examples are closer the the boundary. Their approach relies on the size of the initial dataset to choose between selecting harder or easier examples. The author showed that the pruning helps perform and suggests "hints" of exponential scaling. They highlighted that the success of the pruning relies on finding an adequate difficulty metric. In NLU, Attendu & Corbeil (2023) experimented with a dynamic pruning method for finetuning encoder-only models for joint intent and classification tasks. Their approach consists of a periodic evaluation (spaced by a given number of epochs) that uses the EL2N to assign an importance score to each sample, that model is trained on the most important samples. Their results show that their approach can half the data requirement and prune up to 80% if tolerating a 1% drop in performance.

**Curriculum Learning** is a training approach that aims to improve data utilization by identifying an optimal sequence of training examples. It was initially motivated by the work of Elman (1993) which was an early example of combining ideas from cognitive science with machine learning. It showed that restricting resources (i.e., the data and the memory) early in training and gradually expanding them improves the generalization performance of gradient-based learning. The influential work of Bengio et al. (2009) experimented with small neural LMs and found that CL empirically improved generalization performance. However, recent work has shown the approach used by early CL work is not optimal where it relies on simple handcrafted methods (e.g., the sequence length) to build curricula. Further, these curricula are usually static, not affected by the state of the learner during training. Later studies focused on addressing the aforementioned limitation by dynamically, creating the curriculum during training. For example, Kreutzer et al. (2021) used a vanilla EXP3 algorithm (Auer et al., 2002) that uses the model's loss to guide the data selection process (or learn the curriculum) to train a machine translation (MT) method. Xu et al. (2020) introduced a dynamic approach that uses the changes in the loss or negative log-likelihood as a difficulty measure. Their approach estimates a model competency value to help choose the easiest examples first.

## 3 SPACED SCHEDULING TRAINING

In this section, we describe our *Spaced Scheduling Training* (SST) algorithm for general deep learning (DL) training and our proposed implementation for tuning LLMs. Our method adapts existing spaced repetition algorithms widely used by humans to optimize learning and long-term retention, such as SuperMemo (Wozniak & Gorzelańczyk, 1994). It aims to modulate the intervals at which the learner (in our case, a DL model) trains on a specific example based on how well the information is recalled. The scheduling algorithm uses these scores to estimate a *potentially optimal* interval for each example. The increase (or decrease) of the interval lengths (or spacing) calculation depends on the scheduling algorithm. Still, it is generally affected by the score where a low score shortens the interval and the number of successful consecutive reviews that act as a multiplier. From a DL perspective, our method is motivated by optimal compute allocation. Given a fixed computing budget, an *optimal* learning algorithm should use this budget on examples that contribute positively to performance and avoid ones with minimal or negative impact.

### 3.1 CONCEPTS

We classify each training example throughout the learning process in the categories below to either use a given example for training in the current or future iteration or drop it altogether.

**Trivial examples** represent instances that a model performs well on when seen for the first time. They are defined by a score $z_i > z_{min}$ and repetition count $r_i = 0$. This distinction is crucial when tuning an LLM with significant zero-shot performance.

**Learned examples** represent instances that the model trained on successfully over multiple consecutive repetitions $s_i$. They are defined by $s_i > s_t$ and the number of consecutive reviews with a score $z_i > z_{min}$ where $s_t$ is a hyper-parameter.

**Currently difficult examples** represent instances that are hard for the current model but might be learned later in training and are defined by a score $z_i > z_t$.

**Intractable examples** represent instances that a given model cannot learn after multiple attempts, such as ones with complexity beyond the true model's capacity or which are mislabelled; they are, in effect, intractable from the trained model lens. For instance, it might be intractable for an LLM trained only on natural language to generate a Python function whereas a same-sized code model would find it trivial. They are defined by a score $z_i < z_{min}$ and the total repetition count $r_i > s_t$.

**Useful examples** are instances selected for training in a given iteration. They represent the examples with $z_i \leq z_t$, where $z_t$ is the current score threshold based on the current model competency $\kappa$.

Our proposed algorithm uses useful examples at each training iteration, delays the currently difficult examples, and drops all the other examples described above. The goal of this mechanism is to keep the most useful training examples–used to update the model parameters. It tries to mimic human learning behavior, where optimal learning is correlated with choosing the right level of complexity at the right time. The complete algorithm of the dropping process is shown in Algorithm 3.

## 3.2 ALGORITHM

Our SST algorithm shown in Algorithm 1 follows a two-phase process:

**Phase 1: Warm-up.** The model is trained using random sampling on a subset of the dataset $\mathcal{D}$, defined by $\rho_0$, the *warm-up ratio*. The examples are sampled from $\mathcal{D}$ with replacement using SAMPLEWITHREPLACEMENT, randomly or stratified by a data category if available. The algorithm can use any metadata that can cluster the data. For example, using a task ID in a multitask setting, the data language (e.g., English, French, etc.), or the task type (e.g., code generation, reading comprehension, etc.). The stratification ensures the model sees all possible categories during warm-up. In summary, this phase aims to delay the spaced scheduled training until the model starts producing acceptable outputs to get meaningful scores. The positive effect of this stage is further increased when stratified sampling is used

**Phase 2: Spaced Scheduled Training.** At each epoch $e$, the algorithm performs a series of evaluate-train iterations $n$. The evaluation consists of scoring the set of candidate examples $\mathcal{C}_{new}$ to determine whether to use it in the current or future interaction or drop it altogether. The examples are drawn from $D_{u_e}$, where $D_{u_0} = D$, similar to phase 1, but using sampling without replacement instead and $\rho_{new}$ the *new samples ratio*. The algorithm uses COMPUTESCORES to compute a discrete score $z \in \{0, \ldots, z_{max}\}$ for each example (SuperMemo2 uses $z_{max} = 5$), reflecting the quality of the prediction, where $z = z_{max}$ means a perfect output. Then feeds the score to SUPERMEMO (the implementation of SuperMemo2 (Wozniak & Gorzelańczyk, 1994) shown in Algorithm 2) to schedule each example in its respective target iteration $n_t$, or drops the example following the sample dropping procedure described below. The model is trained on the examples with $n_t = n_{current}$ and the previously scheduled examples. The algorithm keeps track of the examples used for training $D_{u_e}$ that will replace the training dataset in the next epoch. Therefore, once an example is dropped, the model never sees it again. The training epoch finishes when the new examples are exhausted and the schedule is empty. SST then sets $D_{u_e} \leftarrow D_{u_{e-1}}$ and restart the process describe above. Throughout all the epochs, SST keeps an updated model competency value $\kappa$ (initialized by $\kappa_0$ for the first epoch) consisting of the scores' running average, used to compute the minimum score threshold $z_t \leftarrow z_{max} - \kappa - 1$ to place the candidates with a score lower than $\kappa$ by 1 in $D_{u_e}$ since they represent examples currently hard for the model.

---

**Algorithm 1** Spaced Scheduling Training algorithm using the SuperMemo2 algorithm defined in Algorithm 2 and drop function defined in Algorithm 3

---

1: **Input:** Model parameters $\theta$, Training data $\mathcal{D}$, Data categories $\Lambda$, Initial competency $\kappa_0$, Warm-up ratio $\rho_0$, New samples ratio $\rho_{new}$, min correct score $z_{min}$, success repetition threshold $s_t$, max training iteration $T$
2: **Phase 1: Warm-up**
3: $\mathcal{D}_0 \leftarrow \text{SAMPLEWITHREPLACEMENT}(\mathcal{D}, \Lambda, \kappa_0)$
4: $\theta \leftarrow \text{TRAINMODEL}(\theta, \mathcal{D}_0)$
5: **Phase 2: SST**
6: $e \leftarrow 1, \kappa \leftarrow \kappa_0, z_{max} = 5, t = 0, \mathcal{P} \leftarrow \emptyset, D_{u_0} = D$               ▷ Initialization
7: $reviews \leftarrow$ Empty dictionary, $schedule \leftarrow$ Empty dictionary
8: $done \leftarrow$ false
9: **for** $e = 1$ **to** $e_{max}$ **do**
10:      $n \leftarrow 1$
11:      **while** true **do**
12:          $\mathcal{C}_{new} \leftarrow \text{SAMPLEWITHOUREPLACEMENT}(D_{u_{e-1}}, \Lambda, \rho_{new})$
13:          **if** $\mathcal{C}_{new} \neq \emptyset$ **then**
14:             $z \leftarrow \text{COMPUTESCORES}(\theta, \mathcal{C}_{new})$
15:             $\kappa \leftarrow \bar{z}, z_t \leftarrow z_{max} - \kappa - 1$           ▷ Minimum score threshold
16:             $\mathcal{C}_{new}, \mathcal{D}_{u_e} \leftarrow \text{DROPEXAMPLES}(\mathcal{C}_{new}, \mathcal{D}_{u_e}, z_{min}, z_t)$
17:          **else if** $schedule[x] = \emptyset \forall x > n$ & $\mathcal{D}_{u_{e-1}} = \emptyset$ **then**
18:             $done \leftarrow$ true                   ▷ Early stopping
19:             break
20:          **else if** $schedule[n] = \emptyset$ & $\exists x > n$ $schedule[x] \neq \emptyset$ **then**
21:             $n \leftarrow n + 1$ continue       ▷ Future iterations contain scheduled examples
22:          **end if**
23:          $\theta \leftarrow \text{TRAINMODEL}(\theta, \mathcal{C}_{new} \cup schedule[e])$
24:          $n \leftarrow n + 1$
25:      **end while**
26:      **if** done = true **then** break
27:      **end if**
28: **end for**
29: **Output:** Trained model parameters $\theta$

---

### 3.3 COMPUTATION OVERHEAD

Since our proposed approach adds an evaluation process during training, it is essential to understand the compute overhead induced by the extra processing. However, under some conditions, the pruning mechanism used by SST can offset the overhead induced by the evaluation. In this section, we provide the equation that governs the efficiency of our approach. Using Algorithm 1 and estimations from Kaplan et al. (2020), we can write the ratio between the evaluation compute $C_e$ and compute required to train (forward and backward pass) a single example $C_s$ as follows (See Appendix A.6 for details):

$$\frac{C_e}{C_s} \leq \frac{\Sigma_{n=0}^{N-1} d_n}{N - \rho_0 - \Sigma_{n=0}^{N-2} d_n} \tag{2}$$

where $d_n$ is the pruning ratio and $N$ the number of training epoch. Empirical results can use this equation and the measured $C_e$ and $d_n$ to determine the efficiency of SST.

### 3.4 USING SST WITH OTHER MODALITIES AND FUTURE WORK

Later in this work, we show that SST improves the performance of LLM IFT on many evaluation benchmarks. However, the proposed method is not limited to LLMs or text data only but instead can be applied to any type of learner or data modality. Applying SST to another model architecture or data type simply involves choosing the right scoring function that maps an output to a discrete score from 0 to 5 that introduces the least amount of compute overhead. We believe that SST will provide

benefits to training models on other modalities since the underlying idea of selecting the right training examples has been shown to successfully work for images (Sorscher et al., 2022). However, we leave the experimentation to validate this hypothesis to future work. Finally, our method can be used with any spaced repetition algorithm to compute the review intervals. One interesting future direction is to try including the model competency as part of the interval calculation and possibly extract the current competency from the model internals (.e.g., activations, or internal representations).

## 4 EXPERIMENTAL SETUP

### 4.1 SST IMPLEMENTATION FOR LLM IFT

Our implementation uses a Python version of the SuperMemo2 [1] spaced repetition algorithm to compute the review intervals. We set the *minimal correct score* to 3, the $\kappa_0 = 2$ and set both $z_{min}$ and $s_t$ to 3following the original SuperMemo work (Wozniak & Gorzelańczyk, 1994). Further, we set $\rho_0 = 15\%$, $\rho_{new} = 10\%$ that we identify with ablation studies. As for the data, we define the data category as the source dataset's name. Finally, we introduced a variation of the edit distance that we call *token edit distance* as a scoring mechanism that we describe below.

**Token Edit Distance** is a variation of the edit distance adapted for a generative model output (i.e., tokens). The original score, also called Levinstein distance, computes the number of single-character edits required to transform one string to another. Instead, our proposed variation computes the edits at the token level since it is the smallest atomic output unit of an LLM. Further, choosing a token-level edit distance aligns well with the next token prediction objective function used to train causal LMs. For faster computation, we use a Python wrapper of a C implementation of the edit distance [2]. The choice of this scoring function was also motivated by the compute overhead the score calculation induces. The full implementation is available in our code repository.

### 4.2 DATASET

We use the Tulu V2[3] IFT dataset collection, the latest version of the one initially introduced by Wang et al. (2023), totaling nine datasets. See Appendix A.4 for additional details on the exact dataset list and how the latest version differs from the one introduced in the original work (Wang et al., 2023). Our method aims to alleviate the need to select the datasets or their composition. Thus, the choice was mainly motivated by finding a collection that contains: (1) the most datasets, (2) a mix of human and model-generated datasets, and (3) recent and widely used datasets. The Tulu collection Wang et al. (2023) covers all these aspects. Further, we create a validation set containing 5% of each dataset in the collection. This validation set is **only** used by the RbF baseline to estimate the model strength during training, which is necessary for its workings.

### 4.3 TRAINING AND EVALUATION

We follow the same training setup including hyper-parameters as Wang et al. (2023) and base our training code and data processing on their publicly available code[4]. We set the maximum sequence length to 4096. All the models in this work except the vanilla LLAMA models are trained for 3 epochs (or equivalent by setting the maximum training iterations) using LoRa (Hu et al., 2021) unless stated otherwise. The complete training configuration can be found on our public repository[5]. All experiments were run on 8 A100 Nvidia GPUs. We also follow the same data processing setup as Wang et al. (2023) and use chatbot-style prompts (i.e., assistant and user) to handle the datasets with more than one turn (e.g., ShareGPT) and system messages when available (e.g., Open-Orca).

For a comprehensive evaluation, we follow the setup proposed by Touvron et al. (2023) where each model is tested on five capabilities: code, commonsense reasoning, word knowledge, reading comprehension, and math. Each capability score represents an average over multiple tasks. Further, the

---

[1]https://github.com/alankan886/SuperMemo2

[2]https://github.com/maxbachmann/python-Levenshtein/

[3]Available in the authors' official code repository: https://github.com/allenai/open-instruct

[4]https://github.com/allenai/open-instruct/

[5]Available after the blind review period.

models are tested on two popular benchmarks: Massive Massive Multitask Language Understanding (MMLU) (Hendrycks et al., 2020) and Big Bench Hard (BBH) (Suzgun et al., 2022). When evaluating the vanilla LLAMA-2 we use the non-chat prompt suggested by the original work (Touvron et al., 2023). It is worth noting Touvron et al. (2023) used proprietary evaluation code and the results we show in this work are reproduced using the BigCode LM-eval-harness for code capability[6] and EleutherAI LM-eval-harness (Gao et al., 2021) [7] for the rest. Therefore, our results might differ from the original work Touvron et al. (2023). Nonetheless, since all the baselines and our approach are evaluated with the same method, the findings of our experiments remain valid.

## 4.4 BASELINES

We train the following baselines to compare and contrast our method (**SST**) to existing ones and other naive baselines. All trained models are based on LLAMA-2 (Touvron et al., 2023) and are trained on the full TULU V2 described above, unless stated otherwise. Further, we train 7B and 13B model sizes for each baseline.

○ **SST**$_{rand}$ (ours): Training using SST with random scores.
○ **LLAMA-2**: The vanilla LLAMA-2 pre-trained model.
○ **RANDOM**: Training using random sampling–The most widely used sampling strategy.
○ **STATIC**$_{ppl}$: Training on a dataset pruned offline using perplexity Marion et al. (2023), using the vanilla model as reference model.
○ **RbF**: Training using Repeat before Forgetting (RbF) (Amiri et al., 2017) (See A.5.2).
○ **DATA DIET**: Training using our adaptation of the pruning algorithm proposed by Attendu & Corbeil (2023) for sequence outputs (See A.5.1).

## 5 RESULTS

### 5.1 SPACED SCHEDULING TRAINING PERFORMANCE

Table 1 shows the results of our main experiment highlighting the performance improvements of SST compared to baselines described above on the set of capabilities and benchmarks described in the evaluation section.

Table 1: Performance of Spaced Scheduling on LLM capabilities. We report the performance difference between each tuned model and the base pre-trained model of the same size in parentheses.

| Size | Model | Code | Commonsense Reasoning | World Knowledge | Reading Comprehension | Math | MMLU | BBH |
|------|-------|------|----------------------|-----------------|----------------------|------|------|-----|
| 7B | LLAMA-2 | 16.8 | 64.8 | **63.2** | 67.2 | 8.6 | **42.9** | 35.6 |
| | STATIC$_{ppl}$ | 18.1 | 65.8 | 60.1 | 67.0 | 15.8 | 41.2 | 34.3 |
| | RANDOM | 18.7 | 65.4 | 60.8 | 67.2 | 20.8 | 42.1 | 35.9 |
| | DATA DIET | 17.0 | 64.2 | 60.3 | 66.9 | 7.1 | 42.3 | 33.1 |
| | RBF | 13.4 | 61.2 | 60.1 | 64.9 | 5.7 | **42.9** | 32.6 |
| | SST$_{rand}$ | 17.2 | 64.1 | 59.1 | 67.0 | 9.3 | 42.7 | 34.1 |
| | ⋆ SST | **21.2** | **66.0** | 62.7 | **68.0** | **23.9** | 42.2 | **36.8** |
| 13B | LLAMA-2 | 24.5 | 67.1 | **71.8** | **75.9** | 16.3 | **52.9** | 40.7 |
| | STATIC$_{ppl}$ | 27.0 | 67.5 | 74.0 | 73.9 | 29.9 | 52.6 | 42.4 |
| | RANDOM | 29.2 | 67.8 | 73.5 | 73.9 | 29.5 | 52.6 | 42.2 |
| | DATA DIET | 25.9 | 66.1 | 67.3 | 72.6 | 10.6 | 51.9 | 40.0 |
| | RBF | 23.1 | 63.1 | 67.7 | 71.0 | 14.0 | 52.8 | 39.9 |
| | SST$_{rand}$ | 26.1 | 66.5 | 68.4 | 73.0 | 17.9 | 52.8 | 41.3 |
| | ⋆ SST | **32.8** | **68.3** | 70.9 | 74.1 | **31.6** | 52.8 | **43.3** |

SST improves the overall performance (4/7 evaluations) for both model sizes compared to all the baselines. The results show that our approach significantly improves the reasoning capability, as

---

[6]https://github.com/bigcode-project/bigcode-evaluation-harness
[7]https://github.com/EleutherAI/lm-evaluation-harness

demonstrated by improved code, commonsense reasoning, MATH, and BBH performance. When analyzing the schedule followed by SST, we can clearly see that it focuses on examples with short targets (e.g., Flan V2), and then switches to longer ones (e.g., ShareGPT or OpenOrca). Using the findings of Mukherjee et al. (2023), showing that the sentence length correlates with example complexity, we can induce that SST focuses initially on easier examples. However, our qualitative analysis showed that the length is not always an adequate proxy for the sample complexity. For example, both model sizes struggle with world knowledge targets containing a single word (e.g., the model predicts "Miami Beach" instead of "Miami" even after seeing the example more than 3 times). The same behavior occurs for logic or arithmetic targets especially when input context length is short. These findings might explain to low scores on world knowledge and MMLU. When comparing the 7B and 13B variants, we noticed that the transition to longer examples happens earlier for the 13B model showing that the model size affects the schedule, showing the necessity of using online scheduling and pruning algorithms. Interestingly, $\text{STATIC}_{ppl}$ shows better and random results in more than one category for the 13B model, matching the finding of Marion et al. (2023) that showed that larger models perform better as a reference model suggesting that perplexity might useful as online pruning metric. However, we leave this future work. The results show that RBF performs poorly even compared to the vanilla model. Our analysis showed that this is due to the scaling of the loss value introduced by IFT since it is performed in a multitask learning (MTL) setting with model targets varying in lengths (e.g., Flan V2 and ShareGPT have an average target length of 30 tokens and 350 tokens, respectively). The scaling affects RBF's performance since it uses the example loss to compute the schedule. This issue is even more amplified since this method uses the average validation loss to estimate the current model strength used to calculate the schedule. Therefore, this measure also suffers from the loss scaling. Further, when using RBF, the composition of the validation set becomes crucial since it needs to provide a good overview of the model strength on each task in the MTL setup and the simple stratified sampling by dataset we used to create the validation set might not be enough. However, it is worth noting that original work Amiri et al. (2017) was intended for simple classification tasks in a single-task learning setting which explains why RBF is not suitable for the IFT setup we are interested in. DATA DIET also shows a poor performance, where the pruning process removes valuable examples at every epoch, preventing the model from seeing them throughout the training process. We performed a quick experiment where we forced the examples that SST deemed valuable but not DATA DIET and we noticed a performance improvement, pointing to the pruning metric. However, the reason might be related to the EL2N adaptation to sequence data since the original work was intended for a classification task. The work of Marion et al. (2023) showed a similar behavior where the pruning of pre-train data where the data format and learning objective is similar to IFT performs worse than random pruning. To ensure the statistical significance of our results, we performed a paired $t$-test of SST and $\text{SST}_{random}$ for GSM8k and MATH results in every category[8]. We found that the 95% confidence intervals (CI) for the performance difference in favor of SST are $(0.140 - 0.0312)$ and $(0.020 - 0.0294)$ for the 7B and 13B models for the MATH, and found the 95% CI are $(0.024 - 0.044)$ and $(0.020 - 0.029)$ for the 7B and 13B models for GSM8K, respectively.

When using the token edit score, the compute required to evaluate a single example can be estimated as $C_e = C_f + C_{tds} + \epsilon$, where $C_f$ represents the compute required to perform a forward pass, $C_{tds}$ is compute required to get a score using the token edit distance, and $\epsilon$ is extra compute used call the super memo algorithm. For a 7B+ LLM, we can safely assume for the given implementation that $C_f >> C_{tds} + \epsilon$ since the target is at most 400 tokens. Thus $C_e = C_f$. Using equation 2, we can write $C_e/C_s = C_f/3C_f = 1/3$. Knowing that the 7B variant yielded $\Sigma_0^{3-1} d_n = 0.16 + 0.3 + 0.33$ (3 is the number of epochs), we can compute that for this experiment, SST is as efficient as random sampling ($0.3333 < 0.3361$). However, the 13B variant with $\Sigma_0^{3-1} d_n = 0.15 + 0.25 + 0.26$ is considerably more efficient with $\Sigma_0^{3-1} d_n = 0.19 + 0.37 + 0.40$.

Finally, we performed an ablation study to show the benefit of each main component of our SST algorithm. We show the results in Table 2 on the GSM8K (Hendrycks et al., 2021) and MATH (Cobbe et al., 2021) tasks (represent the tasks in the math capability used by (Touvron et al., 2023)) using our 7B variant as they require enhanced reasoning ability that shows cases one of the main benefits of our approach.

---

[8]For MATH we used the original categories (e.g., algebra, pre-calculus), and for GSM8K we clustered the examples into 10 equal-size bucket based on example length

The results show that each component contributes positively to the performance increasing the GSM8k test score by 7.1 points and MATH by 5.8.

## 5.2 SPACED SCHEDULING REDUCES CATASTROPHIC FORGETTING

Table 2: Effect of each component of the SST algorithm using LLAMA-2 7B

| Method | GSM8k | MATH |
|---|---|---|
| Space Scheduling | 32.9 | 1.9 |
| + Stratified warm-up | 34.5 | 2.4 |
| + Example dropping | 37.8 | 5.6 |
| + Competency threshold | **40.0** | **7.7** |

While IFT improves the instruction-following performance of LLMs, studies (Wang et al., 2023) have shown that it can lead to catastrophic forgetting where the tuning process erases the knowledge acquired during pre-training. The results in Table 1 show that in all cases where the performance of the vanilla pre-trained model performs better than the tuned models, Spaced Scheduling reduces the gap of performance by an average of 62% on the tested tasks. We hypothesize that our sample-dropping mechanism makes the learning process focus on useful and learnable examples. However, we leave further analysis of how Space Scheduling conserves valuable internal representation for future work. Further, the results show that our method also improves the performance on most of the test capabilities and benchmarks.

## 6 DISCUSSION, CONCLUSIONS, AND LIMITATIONS

Our work shows that scheduling and modulating the complexity during the IFT process yields better performance, particularly in reasoning ability. This implies that an *optimal* scheduling might exist when training LLMs and that DL training can still benefit from insight into how humans learn, prompting future work. Another interesting future direction is to better understand why reasoning benefits from the scheduling effect using the model's intrinsic characteristics, such as a probe-based analysis of the learned representations. We show that dropping intractable samples is also beneficial since the model capacity puts an upper bound on the number and complexity of skills the model can learn. For example, Wei et al. (2022) has shown that the ability to generate complex CoT reasoning is a property that emerges with the model size. Therefore, our dropping mechanism saves compute budget since in this example a small model will not be able to generate the desired CoT regardless of the number of training epochs. This mechanism is in contrast with traditional Active Learning work (Lewis, 1995), where the hardest example is prioritized. It can also avoid using mislabeled examples, especially with the number of model-generated datasets that are not curated which affects the optimization process as highlighted by Sorscher et al. (2022). The dropping mechanism is also tailored to the model being trained. For example, Figure 1 shows a case where a model trained on Python code could find the example trivial as opposed to a model trained on language only such as LLAMA-2. Finally, we would like to highlight some limitations of our proposed approach that we believe will spark future work. First, the evaluation process that is performed before every learning epoch can induce an important compute overhead. In our experimentation, this overhead is offset by the number of training samples we dropped (e.g., 37.5% for the 13B model). However, this amount is a function of the quality of the training data where in the extreme case all the data can be useful, making the evaluation step pointless. Second, we noticed an interplay between the warm-up phase of SST and learning rate scheduling and we disabled the latter for all our experiments. We only tested our approach on models with at most 13B parameters and it is not clear if larger models such as LLAMA-2 70B would benefit from such an approach. Lastly, we dealing with a large set of IFT datasets, and the issue of the negative impact of some datasets arises. We believe that including an evaluation signal in the dropping mechanism might alleviate this issue but we leave such experimentation for future work.

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

# A  APPENDIX

## A.1  SUPERMEMO ALGORITHM

---
**Algorithm 2** SuperMemo SM2 Algorithm Wozniak & Gorzelańczyk (1994)
---
1: **function** SUPERMEMO($review$, $z$)
2:     $review.repetitions \leftarrow review.repetitions$
3:     **if** $z \geq 3$ **then**
4:         **if** $review.success\_success\_repetitions = 0$ **then**
5:             $review.interval \leftarrow 1$
6:         **else if** $review.success\_repetitions = 1$ **then**
7:             $review.interval \leftarrow 6$
8:         **else**
9:             $review.interval \leftarrow review.interval \times review.ease$
10:         **end if**
11:         $review.success\_repetitions \leftarrow review.success\_repetitions + 1$
12:     **else**
13:         $review.success\_repetitions \leftarrow 0$
14:         $review.interval \leftarrow 1$
15:     **end if**
16:     $review.ease \leftarrow review.ease + 0.1 - (5 - z) \times (0.08 + (5 - z) \times 0.02)$
17:     **return** $review$
18: **end function**
---

## A.2  DROPPING MECHANISM

---
**Algorithm 3** SST dropping Algorithm
---
1: **function** DROPEXAMPLES($\mathcal{C}_{new}$, $\mathcal{D}_{u_e}$, $z_{min}$, $z_t$)
2:     **for** $i, c$ in Enumerate($\mathcal{C}_{new}$) **do**
3:         $s_i \leftarrow reviews[c].success\_repetitions$
4:         $r_i \leftarrow reviews[c].repetitions$                              $\triangleright$ Dropping prorecess
5:         **if** $c \notin reviews$ & $z_i > z_{min}$ **then**
6:             $\mathcal{C}_{new} \leftarrow \mathcal{C}_{new} - c$                        $\triangleright$ Trivial example
7:             continue
8:         **else if** $c \in reviews$ & $(z_i < z_{min}$ & $r_i > s_t))$ **then**
9:             $\mathcal{C}_{new} \leftarrow \mathcal{C}_{new} - c$                     $\triangleright$ Intractable examples
10:             continue
11:         **else if** $c \in reviews$ & $(z_i > z_{min}$ & $s_i > s_t))$ **then**
12:             $\mathcal{C}_{new} \leftarrow \mathcal{C}_{new} - c$                      $\triangleright$ Learned examples
13:             continue
14:         **else if** $z_i < z_t$ **then**
15:             $\mathcal{C}_{new} \leftarrow \mathcal{C}_{new} - c, \mathcal{D}_{u_e} \leftarrow \mathcal{D}_{u_e} + c$       $\triangleright$ Currently difficult example
16:             continue
17:         **end if**
18:     **end for**
19:     **return** $\mathcal{C}_{new}, \mathcal{D}_{u_e}$
20: **end function**
---

## A.3  IFT FOLLOWS A CURRICULUM

One of the main underlying ideas of our approach is that optimal IFT follows a curriculum. To evaluate this hypothesis we compared the results of a LLAMA 2 7B trained using our method to three variations:

○ **Variation 1**: Select the hardest examples beyond the model competency at every training step.
○ **Variation 2**: Like Variation 1 for the first half of the training, then uses SST for the second half.

○ **Variation 3**: Similar to Variation 2, but uses random sampling in the second half of training.

Here, the models are tuned on Tulu V2-Medium for a total of 5k training samples. We evaluate the models on the GSM8k and MATH datasets since they show this phenomenon better due to the low zero-shot on these tasks. The results of this experiment are shown in Table 3.

The results of Variation 1 show that the performance drops drastically when selecting the most

Table 3: Effect of examples difficulty scheduling on performance.

| Method | GSM8k | MATH |
|---|---|---|
| Spaced Scheduling | **36.7** | **5.9** |
| Variation 1 | 27.2 | 3.1 |
| Variation 2 | 32.6 | 3.9 |
| Variation 3 | 31.8 | 3.3 |

difficult examples at each training step, showing that the difficulty order affects performance, suggesting that there exists an optimal curriculum for IFT or LLM learning in general. This is different from traditional Active Learning (Lewis, 1995; Zhu et al., 2008) work where the most difficult examples (e.g., examples with high entropy) as selected first. Our findings are similar to the one from a recent study (Mukherjee et al., 2023), where they show that a LLaMa 13B model performs better when it is trained on easy examples generated from ChatGPT followed by ones generated by GPT-4 that contain longer, more elaborate reasoning and CoT examples. Further, the results of variations 2 and 3 show that using a non-optimal schedule early in training can also be non-reversible acting similar to a bad model initialization. Both random sampling and Spaced Scheduling were not able to recover the performance. However, our approach reduces the performance gap better.

## A.4 DATASET

We use the Tulu V2 dataset collection available in the authors' code repository. At the moment of releasing our work, the latest version of the Tulu V2 collection[9] contains the following modifications on top the initial version introduced by Wang et al. (2023):

○ Adds **Open-Orca** (Lian et al., 2023): Augmented version of FLAN V2 (Longpre et al., 2023) [10].
○ Adds **Evolve-Instruct V2** (Xu et al., 2023a): An augmented version of Alpaca (Taori et al., 2023).
○ Adds **LIMA** (Zhou et al., 2023): A collection of 1k highly curated examples from various sources.
○ Down sampled **FLAN V2** (Longpre et al., 2023) to 50k examples.
○ Removes **Alpaca** Taori et al. (2023).
○ Removes **Dolly** Conover et al. (2023).

## A.5 BASELINES IMPLEMENTATION DETAILS

### A.5.1 DATA DIET ALGORITHM

During our evaluation, we compared the performance of our algorithm against the Data Diet approached introduced by Attendu & Corbeil (2023). This method used the EL2N to compute an importance value used to prune the data. The original implementation was introduced for the joint intent classification and slot extraction. However, due to the generative nature of the tasks used in work, we adapt this algorithm following Marion et al. (2023), in which the authors define EL2N score for text sequences as the L2 norm of the error vector as follows:

$$\text{EL2N}(z_i) = \frac{1}{t}\Sigma_i^t \parallel \hat{y}_t - y_t \parallel_2 \tag{3}$$

### A.5.2 RbF ALGORITHM

We use the best-performing hyper-parameters when evaluating using the RbF algorithm as shown in the original work Amiri et al. (2017). Precisely, we set the recall confidence value $\eta = 0.5$ and use the Cosine scheduler. Further, we use a validation set that contains examples from all the datasets which is used to estimate the model strength during training.

---

[9]https://github.com/allenai/open-instruct/commit/4a2e9dd0c1236f8cc0cdd15e5316e7c371a16624
[10]This dataset is the open implementation of Orca (Mukherjee et al., 2023)

> **Instruction** : Reverse a linked list.
>
> **Output**: def reverse(head): prev = None cur = head while cur: next = cur.next cur.next = prev prev = cur cur = next head = prev return head".

Figure 1: An ambiguous training example from Baize dataset (Xu et al., 2023b), filtered by our dropping mechanism. Note also that our base model was also not trained extensively on code.

### A.6 COMPUTATION OVERHEAD

In this section, we provide the equations that describe the compute requirement of our proposed SST algorithm. Using the equations below we can provide efficiency bounds for our method.

The compute $C_r$ required to train a transformer-based model on a dataset $D$ for a $N$ epochs using random sampling can be written as:

$$C_r \approx N|D|C_s \approx NC_{r_0} \tag{4}$$

Where $C_s \approx 3C_f$ is the compute required to train on a single example, and $C_{r_0}$ is the compute required for a single epoch. $C_f$ is the compute required for a forward pass. Here, we are using the approximation that $C_b \approx 2C_f$, where $C_b$ is the backward pass compute requirement, following Kaplan et al. (2020).

Using SST, the compute requirement $C_{SST}$ can be written as:

$$C_{SST} = C_{SST_0} + \Sigma_{n=1}^{N-1} C_{SST_n} \tag{5}$$

$$C_{SST_0} = (\rho_0 C_r) + ((1 - \rho_0)|D|C_e + (1 - d_0)C_{r_0}) \tag{6}$$

$$C_{SST_n} = ((1 - d_{n-1})|D|C_e + (1 - d_n)C_{r_0}) \tag{7}$$

where $C_{SST_0}$ and $C_{SST_n}$ are the compute requirements of the first epoch and the subsequent epochs, respectively. For the first epoch, the compute requirement (first term) consists of the warm-up on a subset of the dataset defined by $\rho_0$, followed by an evaluation and train interaction (second term) where $C_e$ is the compute required to evaluate and score a single example and $d_0$ is the ratio of dropped example during the evaluation process. Here, the model is trained on the remaining examples $1 - d_0$ with a cost equivalent to training using random sampling $C_{r_0}$. After the first epoch, the evaluation is performed only on non-dropped examples of the previous epoch. Therefore, we can write:

$$C_{SST} = (\rho_0 C_{r_0}) + ((1 - \rho_0)|D|C_e + (1 - d_0)C_{r_0}) + \Sigma_{n=1}^{N-1}((1 - d_{n-1})|D|C_e + (1 - d_n)C_{r_0}) \tag{8}$$

Compared to random sampling, $C_{SST}$ contains an overhead induced by the evaluation process $C_{overhead}$, and the compute saved by dropping examples $C_{drop}$ (skipping training). Using Equation 8, we summarize both quantities as follows:

$$C_{overhead} = (1 - \rho_0)|D|C_e + |D|C_e \Sigma_{n=1}^{N-1}(1 - d_{n-1}) \tag{9}$$

$$C_{drop} = d_0 C_{r_0} + \Sigma_{n=1}^{N-1} d_n C_{r_0} = C_{r_0} \Sigma_{n=0}^{N-1} d_n = |D|C_e \Sigma_{n=0}^{N-1} d_n \tag{10}$$

In order for SST to avoid inducing any additional compute overhead when compared to random sampling, $C_{drop}$ must be greater or equal to $C_{overhead}$. The latter depends on two quantities: (1) $C_e$, the cost to evaluate and compute a single example's score, and $d_n$, the ratio of dropped examples at each epoch. One way to study the relationship between $C_{drop}$ and $C_{overhead}$ is to quantify $C_e/C_s$,

the ratio between the cost to evaluate and train a single example. This ratio can be written as follows based on Equations 9 and 10:

$$\frac{C_e}{C_s} \leq \frac{\Sigma_{n=0}^{N-1} d_n}{N - \rho_0 - \Sigma_{n=0}^{N-2} d_n} \tag{11}$$

Using Equation 11, we can obtain an upper bound on the ratio $C_e/C_r$ by using the best-case scenario where SST drops all the examples (the case where a model is fully trained or over-fitted on the data), that is $d_n = 1 \forall n$. Using Equation 11, we find

$$\frac{C_e}{C_s} \leq \frac{N}{1 - \rho_0} \tag{12}$$

