# OpenReview forum: "Spaced Scheduling Enhances Instruction-Prompted Reasoning in Large Language Models"
_ICLR.cc/2024/Conference — Submitted to ICLR 2024_

### Official Review · Reviewer_QXyK · 2023-10-27

**Soundness:** 2 fair
**Presentation:** 3 good
**Contribution:** 2 fair
**Rating:** 5
**Confidence:** 4

**Summary:**

The authors argue that spaced scheduling (inspired by spaced repetition, a popular technique used by students to memorize content)
can be used to better instruction-tune LLMs. They aggregate existing
instruction tuning datasets into one training benchmark, Mercury-instruct.
They then verify, on this benchmark, that their technique outperforms
random data selection when instruction tuning LLAMA-2 at model sizes 7B and 13B.

**Strengths:**

1. Motivating the technique with relevant psychology literature.

1. Applying the technique to instruction tuning LLMs, which is a research
  topic that is attracting considerable attention.

1. Conducting an ablation analysis on the components of the proposed algorithm.

**Weaknesses:**

* Discussion of related work. For example, spaced scheduling for deep
  learning has been considered in **Hadi Amiri et. al**:
  *Repeat before Forgetting: Spaced Repetition for Efficient and Effective
  Training of Neural Networks (ACL 2017, see page 2404)*

* The proposed method does not appear to be motivated with a memory model; compare
  **Amiri et al.** or **https://arxiv.org/pdf/1602.07032.pdf**, both works seem to motivate
  their proposals based on a memory model.

* In my opinion, the empirical part should have at least a comparison to
  another spaced scheduling method (compare **Amiri et al.**).

* In my opinion, it is hard to conclude if one should use the proposed method
  or some other online scheduling approach. For example, there is prior
  relevant work on automated curriculum learning, see for example **Kreutzer et. al**: *Bandits Don’t Follow Rules: Balancing Multi-Facet Machine Translation with Multi-Armed Bandits (ACL 2021)*. While I am **not necessarily** advocating direct comparison
  to the algorithm of **Kreutzer et al.**, I think the empirical part would be
  more solid by having a comparison to one or two additional approaches
  that schedule the data dynamically.

**Questions:**

My initial rating / recommendation is inclined towards rejection because:
* the novelty claim needs a better positioning wrt. previous work
* the empirical investigation feels limited.

I am leaving some questions that would greatly help me to improve
my assessment and in case change the rating / recommendation towards acceptance.

**Major Questions**
1. Could you position your work wrt. to **Amiri et al.**? What makes this proposal
of spaced scheduling novel wrt. prior work?

1. Could you elaborate on why there is not a comparison to another spaced
  scheduler or to other approaches that dynamically schedule the training examples? (e.g. **Kreutzer et al.**)

**Minor Questions**
* Table 1: The performance decrease on some tasks (e.g. World Knowledge)
might be due to instruction tuning on the Mercury dataset. Since one
is interested in comparing instruction-tuning strategies, it might be
worth considering using the **LLAMA-2 Mercury**, which is instruction-tuned without spaced scheduling, as the baseline and reporting the gains/losses wrt. to **LLAMA-2 Mercury**.

---

> ### Author Response · Authors · 2023-11-22
> **Thank you for your feedback**
>
> First, we would like to thank you for the very valuable and relevant suggestion that directed the changes that needed to be done as part of this rebuttal, improving the paper. Amiri et. al, escaped our radar (spaced scheduling is in the title!!).
>
> - **Re: Discussion of related work** Based on your comment, we have reworked the entire related work and introduction section as we can't claim novelty on using Spaced Repetition in DL.
>
> - **Re: memory model motivation**: We have added the memory model discussion to both related work and the introduction.
>
> - **Re: Comparison to Amiri et al.**: We have added the suggested comparison. The results are in Table 1, and a detailed analysis is in Section 5.1 on page 8. Also, based on feedback from other reviewers, we've added 4 more baselines (Table 1 and section 4.4 for details), enhancing the demonstration of our work's benefits.
>
> - **Re: position your work**: We have added a full paragraph in the introduction (p2) to position our work wrt to Amiri et al.
>
> - **Re not comparing to other approaches**: The main reasons that we did not initially include a comparison to other CL or pruning work is: (1) most prior work either focuses on basic classification task or MNT. Thus comparing these methods means significant changes to
> the original algorithm since older work uses LSTM (such as in Amiri et al.)  MLP, or encode-only transformer models (Bert, Roberta) which is different from decoder-only model used recently.  For example, the implementation of an entropy-based active learning with an
> encoder + classification head is almost distinct from implementing it for a decoder-only. (2) to the best of our knowledge, none of the existing work addresses large scale multitask  (e.g, Flan v2 has 1k+tasks). (3)  to the best of our knowledge, none of the existing work focuses on IFT with large-scale models (7B+) which present other challenges such as the possibility of erasing
> per-trained knowledge as opposed to training an LSTM from scratch to perform sentiment analysis. **However, we now have 5 baselines for comparison**.
>
> - **Re: Table 1** While that would make the result a bit easier to read, we believe that will hide the reduction of catastrophic forgetting of our approach. However, we have merged the table into one better clarity.
>
> We hope that the major updated we made to the baseline, related work, introduction, and analysis are sufficient to steer our submission toward an acceptance.
>
> Thank you!

---

### Official Review · Reviewer_aAUa · 2023-10-29

**Soundness:** 2 fair
**Presentation:** 2 fair
**Contribution:** 2 fair
**Rating:** 3
**Confidence:** 3

**Summary:**

The paper proposes an adaptive scheduling strategy called spaced scheduling motivated by the spaced repetition learning method used by humans. The approach aims to perform the data mix selection process online during training, tailoring the training data composition to the chosen pre-trained model. In addition, the paper creates a new instruction meta-collection, i.e., Mercury-Instruct.

**Strengths:**

- The introduction seems interesting. The analogy drawn to human learning processes is quite inspiring.
- The author performed an ablation study to show the benefit of each main component of the proposed algorithm.

**Weaknesses:**

- Some parts of the writing are ambiguous. I think it is better to provide a representative concept figure.
- The experimental results are not so good. Stating that there is an overall performance improvement seems risky because there was a performance gain in four out of seven evaluations.
- While there are numerous hyperparameters within the proposed algorithm, the impact of their variations on performance has not been analyzed. Some ablation studies on hyperparameters, e.g., $s_t$ and $\rho_0$, seem necessary. This is necessary to determine whether this method is insensitive to hyperparameters.
- There are often instances where explanations are missing. For instance, it would be advisable for the authors to explain why the formula for the minimum score threshold based on competency is $z_t \leftarrow z_{max} - \kappa -1$.

**Questions:**

- How do you define the minimum score to deem a response good enough and the threshold repetitions to deem an example learned?
- Could you explain in more detail how you define the data categories?
- How do you define initial competency?
- Could you please provide a more detailed explanation of the criteria used to construct the Mercury-Instruct collection?
- Was there no paper using CL in instruction tuning for LLM previously? If there was, please explain the reason for not comparing with them.

---

> ### Author Response · Authors · 2023-11-22
> **Thank you for your feedback**
>
> - **Re: parts of the writing are ambiguous**: We reworked all the methodology section (section 3) to make it as clear as possible, and split Algorithm 1 into multiple parts. While we agree that a figure could be better, the space does not permit especially with the additional baselines and analysis we performed. However, if you still think the update we made a still ambiguous, we will be happy to make the update for the camera-ready version.
> - **Re: the experimental results are not so good:**  The reason we claim the overall performance is better is that our method is better on 4 out of 7 tasks. While that might seem ""risky"" we believe that the method is showing signification improvement, because
>    - When our method improves a particular type of task, the gains are substantial. (e.g., 15 acc points for math)
>    - The values we show in Table 1 are averages across multiple datasets (e.g., reading comprehension contains
> 8 datasets), thus an increase of 1.2 for example is very significant.
>    - On all the benchmarks where our method scored lower than the pre-trained model, our method is always better compared to the random training,  thus the claim that our method improves over random training remains valid.
>    -  On the main capabilities that our method improved (reasoning), we performed a paired t-test to prove the
> statistical significance of our results. Since other reviewers saw our results as strengths  (e.g, Review 1: "Spaced repetition consistently improves accuracy across all 5 benchmarks  and 2 models, demonstrating robust gains.", Reviewer 2: "quantitive results on 4 benchmark  suite show the effectiveness of the proposed methods versus baseline or random mixture methods;"),  please let us know what parts we should clarify, and we will be happy to make the needed updates when we submit the camera-ready version
>
> - **Re: About the hyperparameters**: Thank you for the valuable suggestion we agree that this was indeed missing from the initial submission. We have added an ablation study of the hyperparameters in the appendix."
> - **Re: explanations are missing**: We clarified the score threshold as you suggested in section 3.2, phase 2. Further, we reworked the algorithm section to make sure that all the aspects are clear."
> - **Re: The minimum score value, and initial competency**: We clarified how we set that value  in section 4.1 (TLDR; it is based on SuperMemo default value)"
> - **Re: the data categories**: We rely on any categorization that exists already in the dataset. A simple categorization is to use the dataset ID (since we only need the value for the stratification). Another possibility is to use categorization in the dataset. For example, the open orca dataset has GPT-4 and GPT-3.5 data splits. In such a case, we use the aforementioned categories
>  rather than the dataset ID. We have added this clarification to section 3.2, phase 1.
>
> - **Re: Design choice of selected instruction dataset** Since our proposed method alleviates the need to select the datasets or their composition, we chose a dataset that contains the most used IFT dataset. We agree that this was missing from our dataset section and we have added it to our latest version of the work to section 4.2. However, note that we longer include Mercury as part of our contribution since the Tulu authors have released a V2 of their dataset that includes the dataset we added. Thus, we have
> updated the dataset section to reflect this."
>
> - **Re CL prior work**: "To our knowledge, there is no CL work for IFT. However, we have reworked the related work section
> to add additional studies that intersect with our work."

---

### Official Review · Reviewer_CD6W · 2023-10-31

**Soundness:** 3 good
**Presentation:** 3 good
**Contribution:** 3 good
**Rating:** 5
**Confidence:** 4

**Summary:**

The paper discusses the advancement in instruction tuning for large language models, a method that has propelled their popularity. While most prior work has concentrated on the development or improvement of datasets, instruction tuning has sometimes led to a decline in performance. To address this, researchers have been carefully selecting the best dataset mixes through intensive ablation studies. The paper introduces a new adaptive scheduling strategy, termed "spaced scheduling", inspired by the spaced repetition learning method in humans. This approach dynamically selects the training data mix online, specifically tailored to the pre-trained model, eliminating the need for extensive studies on training data compositions. The results indicate that Spaced Scheduling surpasses random sampling, requires less training data, and prevents catastrophic forgetting. It also delivers balanced performance across all benchmark subcategories.

**Strengths:**

- The paper is well-written and clearly presented;
- The paper tackled the important problem of data mixture & curriculum  learning of instruction tuning for large language model and proposed a novel method, "space scheduling", quantitive results on 4 benchmark suite show the effectiveness of the proposed methods versus baseline or random mixture methods;
- Detailed ablation and qualitative examples w.r.t other scheduling variants have been presented to show the effectiveness of the proposed
- It is great to show the proposed methods are based on LoRA, which offers extra accessibility to large research community;

**Weaknesses:**

- There is a missing comparison in Table 1 versus Tulu as mentioned in 4.2 for the effectiveness of MERCURY versus original Tulu paper. Besides, MT-Bench or other human-involved evaluations might also be good to show the comprehensive effectiveness of the proposed methods;
- Another concern of the proposed method is that it seems the benefits are enlarged for Math/Code with both MERCURY / Space Scheduling. However, when adding OpenOrca dataset only will contribute to that effect should be ablated;
- The scalability of the proposed method is questionable but it is great to show the purposed methods are based on LoRA;

**Questions:**

- Just wondering the performance based on OpenOrca / Tulu-only for 7B/13B models to show the comprehensive view of MERCURY；
- Could the authors imply the decreased performance on MMLU / World-Knowledge, is that due to the introduce of OpenOrca or other datasets may pose negative impact on the overall performance?
- Could the authors explain the design choice of selected instruction dataset more, since according to Tulu, different data mixture as well as each dataset may contribute to positive/negative effects on each domain;

---

> ### Author Response · Authors · 2023-11-22
> **Thank you for your feedback!**
>
> - **Re missing comparison in Table 1 versus Tulu**: The only reason why we don't include comparison is Tulu in Table 1 is the fact that Tulu is based on Llama, and our model is based on  LLama2, which gives an unfair advantage to our method since Llama2 is superior to Llama1.  However, with the dataset changes described in the main comment, we have included a Tulu V2 comparison that
> corresponds to training Llama2 on the tulu dataset."
> - **Re: the scalability of the method**: We've added section 3.3, which shows the math equations that govern the overhead, and complete details of how we derived the equation in Appendix A.6. Further, we added a paragraph (in section 5.1, page 8) in the quantitative results that shows the numerical values, proving that the proposed method is on part with random sampling for 7B and more efficient for the 13B for the tested setup.
> - **About ablating OpenOrca dataset**: We agree that ideally, we should ablate the addition of each dataset. However, due the amount of work we spent adding 5 baselines and reworking the related work and introduction and  we did not have enough time and compute budget to run these experiments. However, we will be happy to include this in the camera-ready version. Further, we believe that you are bringing up a valuable point and we think that supplementing our method with signals from evaluation sets during training might catch such an issue, where the dropping mechanism can remove examples that might degrade performance.
> However, due to the rebuttal time schedule, we were not able to experiment more with this.
>
> - **Re: Design choice of selected instruction dataset** Since our proposed method alleviates the need to select the datasets or their composition, we chose a dataset that contains the most used IFT dataset. We agree that this was missing from our dataset section and we have added it to our latest version of the work, section 4.2. However, note that we longer include Mercury as part of our contribution since the Tulu authors have released a V2 of their dataset that includes the dataset we added. Thus, we have
> updated the dataset section to reflect this.
> - **Re: about the effect open orca** on MMLU, Looking at Table 1 in the Tulu paper, you can notice that Self-instruct is the dataset that affects negatively MMLU. However, we believe that the score drop on tasks that rely on latent knowledge decreases as a function of IFT dataset, that is, there is a point in time that will need to choose between teaching the model a new skill or keeping the world knowledge that might become stale.

---

### Official Review · Reviewer_Q3ZP · 2023-10-31

**Soundness:** 3 good
**Presentation:** 3 good
**Contribution:** 2 fair
**Rating:** 6
**Confidence:** 3

**Summary:**

This work proposes an adaptive strategy for fine-tuning, where the fine-tuning algorithm actively decides which examples are worth training on based on whether they are too "trivial" or "difficult". This technique is inspired by results in psychology demonstrating how human learning benefits from the technique of spaced repetition. Across LLaMa-2 7B and 13B and 5 benchmark scores, the method outperforms vanilla fine-tuning by 0.6-5.8%. Across several ablations, it's found that every component of this method is needed.

**Strengths:**

1. Spaced repetition consistently improves accuracy across all 5 benchmarks and 2 models, demonstrating robust gains.
2. The paper provides an ablation study for three components of the algorithm, and it seems that all 3 improve performance
3. The paper spends time clearly outlining their precise algorithm

**Weaknesses:**

I overall would appreciate better contextualization/analysis of the method to prove that it's improving upon our current understanding/practice of fine-tuning.

1. Baselines: Though it is encouraging that this model improves over vanilla fine-tuning, this is not the only work in improving data quality/curriculum for fine-tuning. In terms of static pruning methods, one naive baseline is to filter sentences with too high or low perplexity (similar to high/low difficulty), as done [for pretraining](https://arxiv.org/abs/2309.04564). In terms of active learning methods, [Data diets](https://arxiv.org/abs/2306.03208) performs a very similar algorithm to the one in this work, dynamically pruning based on a notion of sample importance (the related work of this paper also provides other references). I believe the paper should reflect this in two ways.
    - The related work and contextualization of the current paper make it seem that this problem has not been studied before, which can be misleading. Better contextualizing the work in terms of prior research in this area will help highlight the novel contributions by this paper.
    - Though the results provide an ablation study, it is unclear where prior work lies in this spectrum. Regardless of the motivation, this paper is providing a new method, and its important to contextualize its gains with respect to prior work. Though there are too many baselines to evaluate all, I would appreciate seeing a reasonable baseline to confirm that along some axis, this work pushes along various fine-tuning tradeoffs.
2. Overhead: This algorithm should induce extra time overhead since examples have to be scored, and "currently difficult" examples may have to go through the model multiple times. The authors should report the time taken by both algorithms to evaluate this slowdown so one can evaluate whether this accuracy improvement is worth the additional cost.
3. Connection to spaced repetition in psychology: From reading the paper, it is not clear to me how connected the algorithm is to spaced repetition learning for humans. According to the paper, spaced repetition says that "brains retain information more effectively when we learn in multiple, spread-out sessions". However, the actual algorithm proposed does not do this, and frames example selection under dynamic filtering based on example difficulty. Even if the analysis in Section 5.3 implies that the method is implicitly setting a curriculum, it is not clear to me how this connects to spaced repetition. I wonder what value the psychological motivation provides in the context of this work, and if there is a connection I'm missing in this regard.

I am happy to adjust my score provided some further analysis along these axes.

**Questions:**

1. Is it possible to see the ablation study for all the benchmarks, or is it only possible to report for MMLU and BBH?
2. Is there any ablation result on the importance of (a) doing this dynamically instead of statically with the pretrained model and (b) adding backdropped data into the dataset?

---

> ### Author Response · Authors · 2023-11-22
> **Thank you for your feedback!**
>
> Thank you for your time and valuable feedback, which significantly contributed to enhancing the quality of our work.
>
> Re **1-Baselines**: Thanks for suggesting the naive baseline. We have a perplexity-based naive and the data diet baselines. The results are in Table 1, and a detailed analysis is in Section 5.1 on page 8. Also, based on feedback from other reviewers, we've added 4 more baselines (Table 1 and section 4.4 for details), enhancing the demonstration of our work's benefits."
>
> Re **' related work and contextualization**:  We've revised the related work section and parts of the introduction (2nd paragraph). This clarifies our work's position as an extension of existing CL and pruning research and emphasizes our novel contributions."
>
> Re **2-Overhead**: We've added section 3.3, which shows the math equations that govern the overhead, and complete details of how we derived the equation in Appendix A.6. Further, we added a paragraph (in section 5.1, page 8) in the quantitative results that shows the numerical values,  proving that the proposed method is on part with random sampling for 7B and more efficient for the 13B for the tested setup.
>
> Re **currently difficult examples** note that our dropping mechanism removes these as they become intractable examples if the model is
> failing to predict them after many attempts properly.
>
> Re: **3 -Connection to spaced repetition in psychology**: We agree that this was not clear in our initial submission. We have updated the related work section (subset Spaced Repetition, 2nd page) to include the work that motivated using the human memory model in DL. We also reflected this in the introduction.
>
> Re: **Question 1** -We already provided the benchmarks' ablation. However, the results were split into two tables. For clarity, Table 1 now contains all the results.
>
> Re: **Question 2- (a)**: Due to the amount of work induced by adding the 4 baselines, we did not have a chance to add it. But we will happily include it in the camera-ready version.
>
> Re: **Question 2- (b)**: The backdropped data is added to a training set in the next epoch. However, we agree that this was not clear in the initial submission. We have updated the algorithm to make it more clear. You can see this in Algorithm 1, line 18; we the
> useful examples used in the previous epoch.
>
> We hope that our major update is sufficient to help adjust your score as you previously suggested.

---

> > ### Comment · Reviewer_Q3ZP · 2023-11-23
> >
> > I commend the authors for their hard work during the rebuttal process. I am especially pleased with the baselines, which really demonstrate how this method pushes beyond current practice in this field. As such, I am changing my score from a 3 to a 6.

---

### Author Response · Authors · 2023-11-22
**Thank you to all the reviewers**

First, we would like to thank all the reviewer for their time and valuable feedback. Also, we want to bring the attention of all the reviewers to the following. We have made the following major updates:

1. **Removal of Mercury Dataset Collection contribution:** The work that we extended earlier added the same datasets and we have decided to remove the contribution to avoid duplication and overlap with the original work.

2. **Addition of More Baselines:** We've significantly expanded our analysis by adding comparisons against **5** additional baselines. This broader comparison will provide a more comprehensive evaluation of our methods.

3. **Updated Related Work:** We have revised the related work and instructions sections to better position our research in the context of spaced repetition, CL,  and pruning techniques.

4. **Compute overhead**:  We've added section 3.3, which shows the math equations that govern the overhead, and complete details of how we derived the equation in Appendix A.6. Further, we added a paragraph (in section 5.1, page 8) in the quantitative results that shows the numerical values, proving that the proposed method is on part with random sampling for 7B and more efficient for the 13B for the tested setup.

Your feedback has been instrumental in these improvements, and we are grateful for your contributions to refining our research.

Thank you!

---

### Meta-Review · Area_Chair_q964 · 2023-12-08

**Metareview:**

This paper considers the problem of deciding how to sample tasks/data when performing instruction fine-tuning of language model. Specifically, it introduces a "spaced scheduling" strategy based on the principle of spaced repetition, using heuristics that measure whether a given example has already been learned and/or is learnable. The proposed sampling strategy provides modest gains in terms of downstream performance when applied to a preexisting model/dataset mixture setting. Reviewers raised issues with a lack of comparison to existing data pruning/selection algorithms and the small/inconsistent gains produced by the method. While the authors included some additional baselines and clarifications in the rebuttal, the combination of modest gains and missing baselines means that this paper should not be accepted this round. The paper should undergo additional revisions and be resubmitted.

**Justification For Why Not Higher Score:**

The paper needs to include a broader set of preexisting baselines and exhibit more convincing improvements to be accepted.

**Justification For Why Not Lower Score:**

N/A

---

### Decision · Program_Chairs · 2024-01-16

Reject